# Galectin 3 (*LGALS3*) Gene Polymorphisms Are Associated with Biochemical Parameters and Primary Disease in Patients with End-Stage Renal Disease in Serbian Population

**DOI:** 10.3390/jcm11133874

**Published:** 2022-07-04

**Authors:** Zoran Kovacevic, Tatjana Lazarevic, Nela Maksimovic, Milka Grk, Vladislav Volarevic, Marina Gazdic Jankovic, Svetlana Djukic, Katarina Janicijevic, Marina Miletic Kovacevic, Biljana Ljujic

**Affiliations:** 1Clinic of Nephrology, University Clinical Center, 34000 Kragujevac, Serbia; kovacevic.zoran@icloud.com (Z.K.); tatjanalazarevickg@gmail.com (T.L.); 2Faculty of Medical Sciences, Department of Internal Medicine, University of Kragujevac, 34000 Kragujevac, Serbia; drsvetlanadjukic@gmail.com; 3Faculty of Medicine, Institute of Human Genetics, University of Belgrade, 11000 Belgrade, Serbia; nelamaksimovic@gmail.com (N.M.); milkagrk@gmail.com (M.G.); 4Faculty of Medical Sciences, Department of Microbiology and Immunology, University of Kragujevac, 34000 Kragujevac, Serbia; drvolarevic@yahoo.com; 5Faculty of Medical Sciences, Department of Genetics, University of Kragujevac, 34000 Kragujevac, Serbia; marinagazdic87@gmail.com (M.G.J.); bljujic74@gmail.com (B.L.); 6Hematology Clinic, University Clinical Center, 34000 Kragujevac, Serbia; 7Faculty of Medical Sciences, Department of Social Medicine, University of Kragujevac, 34000 Kragujevac, Serbia; kaja.andreja@yahoo.com; 8Faculty of Medical Sciences, Department of Histology and Embryology, University of Kragujevac, 34000 Kragujevac, Serbia

**Keywords:** galectin 3, polymorphisms, end-stage renal disease, diabetes mellitus, arterial hypertension

## Abstract

Galectin 3 plays a significant role in the development of chronic renal failure, particularly end-stage renal disease (ESRD). The aim of our study was to investigate the association between Gal-3 and biochemical parameters and primary disease in ESRD patients, by exploring the polymorphisms *LGALS3* rs4644, rs4652, and rs11125. A total of 108 ESRD patients and 38 healthy controls were enrolled in the study. Genotyping of *LGALS3* gene rs4644, rs4652, and rs11125 polymorphisms was performed by polymerase chain reaction–restriction fragment length polymorphism (PCR–RFLP). By multivariate logistic regression analysis, we found that *LGALS3* rs4644 CC and rs4652 AA genotypes were significantly associated with a higher risk for lower hemoglobin, higher level of parathyroid hormone, and also occurrence of diabetes mellitus and arterial hypertension. The CAA haplotype was significantly more common in patients with diabetes, low hemoglobin level, and normal PTH level. It has been observed as well that the ACT haplotype was more common in patients with low glomerular filtration, low PTH, and normal hemoglobin level. We found that the *LGALS3* rs4644 and rs4652 gene polymorphism may be involved in the pathogenesis and appearance of complications in ESRD patients and thus could be considered a new genetic risk factor in this population.

## 1. Introduction

Chronic kidney disease (CKD) is a major health problem worldwide. The last stage of CKD is end-stage renal disease (ESRD), which is characterized by a progressive decrease in glomerular filtration rate (GFR). End-stage renal failure is often complicated by anemia, uremic cardiomyopathy, and renal osteodystrophy, which can be fatal [1]. Therefore, it is necessary to identify the risk factors that can lead to ESRD or death. In addition to well-defined early markers such as diabetes, hypertension, and obesity, it is necessary to identify other markers that would help in the prediction of the disease. It has been shown that genetic factors also have an influence on renal function, pathogenesis, and disease progression, which is confirmed by the results of genome-wide association studies (GWASs). The results of GWASs show that glomerular filtration rate and renal function are affected by several gene loci [2].

Galectin 3 (Gal-3) is the only representative chimera type of the galectin family, which forms a pentameric structure on the cell surface after binding to glycoproteins or glycolipids [3]. The results of numerous studies have shown that Gal-3 plays a significant role in fibrosis, inflammation, and proliferation [4,5,6]. Increased circulating levels of Gal-3 have been associated with various diseases, including cancer, immunological disorders, and cardiovascular disease [7]. There are two single-nucleotide polymorphism (SNP) sites located at chromosome 14 in exon 3 of *LGALS3* which are most common, marked as rs4644 and rs4652 variants. The variant of rs4644 + 191 C > A substitutes histidine to proline at residue 64, whereas the variant of rs4652 + 292 A > C changes threonine at residue 98 to proline [8]. The results of the study showed the bilateral roles of different variants of Gal-3 in different types of malignancies. Previous studies concluded that genetic variants at two Gal-3 gene single-nucleotide polymorphism (SNP) sites are able to change the protein levels of Gal-3 [7]. In patients with acute heart failure, a single-time-point-based measurement of plasma Gal-3 concentration and the severity of myocardial fibrosis are both predictors for the composite outcome of all-cause mortality and rehospitalization at 1-year follow-up. However, no association is found between the polymorphisms of the Gal-3-coding gene (rs4644 and rs4652), plasma Gal-3 concentration, and the degree of myocardial fibrosis [9]. Contrarily, the genotype of rs4644 might be associated with lower left ventricular ejection fraction in patients with dilated cardiomyopathy [10].

Gal-3 plays a significant role in acute renal failure as well as in the development of chronic renal failure. Animal model studies have suggested that Gal-3 expressions were markedly upregulated in both ischemic and toxic types in acute renal failure, and play an important role in acute tubular injury and the following regeneration stage [11]. The results of the meta-analysis suggest that high levels of Gal-3 may increase the risk of all-cause mortality and cardiovascular events in patients with CKD; however, it is probably not a sensitive biomarker for outcomes in hemodialysis patients [12]. Additionally, the results of a clinical study showed that plasma Gal-3 levels were associated with renal insufficiency and poorer survival in patients with chronic systolic heart failure, but they did not record a relationship between Gal-3 and echocardiographic or hemodynamic indices [13]. Additionally, higher levels of Gal-3 correlate with an increased risk of CKD incidents in the general population and are strongly associated with a rapid loss of renal function during 10-year follow-up. No association was observed between the albuminuria level and the serum Gal-3 level [14]. These data suggest that Gal-3 may identify individuals at risk for the development of CKD many years before clinical onset. There are also several small studies that have examined the role of Gal-3 in patients with end-stage renal disease. A study conducted by Mejires et al. showed that urinary Gal-3 levels were not increased in patients with heart failure, despite significantly increased plasma Gal-3 levels [15]. In the multivariable Cox proportional hazard model, Gal-3 levels above the cut-off value remained an independent predictor of all-cause mortality, suggesting that Gal-3 is an independent predictor of mortality in hemodialysis patients [16].

The Gal-3 level has been considered as relevant to various disease treatments by some studies because they play an important role in inflammation and fibrosis process. However, the relationship between the SNPs in the Gal-3 gene and the phenotypic variations in CKD has not been evaluated. However, there is currently no study specifically examining the correlation of Gal-3 gene polymorphisms with the risk and prognosis of ESRD. Therefore, the present study intends to explore the relationship between Gal-3 and biochemical parameters and primary disease in ESRD patients by exploring the polymorphisms *LGALS3* rs4644, rs4652, and rs11125. This may reveal a new aspect of prognosis and treatment in the future.

## 2. Material and Methods

### 2.1. Ethics Statement

All individuals participating in the study provided written informed consent and blood samples were collected under protocols approved by The Ethics Committee of the Clinical Centre of Kragujevac, Serbia (No. 01/20-859, Date 4 December 2020). The study included patients with chronic kidney disease (CKD) from central Serbia, diagnosed and treated at the Clinic of Nephrology and Center of Hemodialysis, Clinical Center of Kragujevac, Serbia. In addition to the genotyping studies, peripheral blood samples were also used to determine standard biochemical parameters relevant for CKD. Furthermore, blood samples from healthy volunteers were also collected. The study was performed in accordance with the Declaration of Helsinki.

### 2.2. Study Population

The study included 108 patients with end-stage renal disease (ESRD) and 38 healthy volunteers matched for age and sex. All patients had a reduced glomerular filtration rate (GFR <10 mL/min/1.73 m^2^). In total, there were 58.3% men and 41.7% women suffering from ESRD. Healthy controls were 73.7% men and 26.3% women. 

### 2.3. Molecular Genetics Methods

Molecular genetic analyses were performed at the Institute of Human Genetics, Faculty of Medicine, University of Belgrade. Genomic DNA was isolated from 5 mL of patients’ peripheral blood with solution-based DNA extraction methods using salting out [17]. Genotypes of *LGALS3* gene rs4644, rs4652, and rs11125 polymorphisms for each patient were determined by real-time PCR method, using TaqMan probes (TaqMan^®^ SNP Genotyping Assays), C___7593635_1_, C___7593636_30, and C___7593637_10, respectively. All reactions were conducted in Real-time PCR machine ABI7500 Real-time system (Applied Biosystems, Foster, CA, USA)

### 2.4. Statistical Analysis

For the comparison of means of the different clinical parameters, between cases and controls, the Mann–Whitney test was used. For the analysis of the pathologies associated with CKD, the Fisher test was performed. The observed genotype frequencies in controls were tested for Hardy–Weinberg equilibrium using the chi-square test. Odds ratios (OR) and 95% confidence intervals (95% CI) for associations between genotypes and CKD, associated phenotypes, and clinical parameters converted to binary variables were estimated by logistic regression while linear regression was used for continuous variables. The analyses were carried out considering two models, one without adjustment and a second adjusting for age and gender. Statistical significance was determined by a *p*-value lower than 0.05. Statistical analyses were performed in SPSS program version 16.0 (SPSS Inc, Chicago, IL, USA), using appropriate statistical methods. Haplotype analysis was performed using HaploView 4.2 software. Haplotype blocks were assessed by the “Confidence Intervals LD” method [18].

## 3. Results

### 3.1. General Characteristic

The demographic, clinical, and biochemical characteristics of the studied patients are summarized in Table 1. The mean age of the ESRD patients (63 males, 45 females) included was 63.05 ± 11.74 years with the mean duration of dialysis at 5.68 ± 5.23 years. The mean age of the 38 healthy controls (28 males, 10 females) was 53.5 ±15.5 years.

### 3.2. Genetic Association Study

Frequencies of alleles and genotypes of the analyzed Gal-3 gene polymorphisms were compared between ESRD patients and controls, and are summarized in Table 2. There was no significant difference between the allele frequency of Gal-3 polymorphisms in the ESRD patients and control subjects (*p* > 0.05). Additionally, haplotype analysis of LGALS3 genes (rs4644, rs4652 and rs11125) was performed. The values of D’ (normalized LD coefficient) and r^2^ (correlation coefficient) are shown in Figure 1. The analysis showed that a haplotypic block was present between the three examined polymorphisms (rs4644, rs4652, and rs11125). The frequency of haplotypes present in patients is shown in Figure 2.

### 3.3. Galectin 3 Gene Polymorphisms Associated with Clinical/Biochemical Parameters

ESRD patients are characterized by a defined biochemical profile acting as a clinical indicator. The obtained results are shown in Table 3. As indicated, three biochemical parameters showed statistical association with the defined Gal-3 gene polymorphisms: hemoglobin, glomerular filtration rate, and parathyroid hormone. Moreover, ESRD patients with *LGALS3* rs4644 CC (*p* = 0.034; OR = 4.83; CI 1.12–20.82) and rs4652 AA (*p* = 0.019; OR =2.86; CI 1.26–6.5) genotypes had 5 and 3 times, respectively, higher risk for lower hemoglobin (˂103 g/L). Additionally, binary logistic regression showed that low glomerular filtration depends on the *LGALS3* rs11125 genotype. Moreover, ESRD patients with the *LGALS3* rs11125 AA genotype have about four-times-higher risk of low filtration (*p* = 0.018; OR = 3.73; CI 1.33–10.5). Furthermore, ESRD patients with *LGALS3* rs4644 CA or AA (*p* = 0.003; OR = 3.52; CI 1.58–7.88) and rs4652 AC or CC (*p* = 0.003; OR = 3.63; CI 1.60–8.23) genotypes had about 3.5 times, respectively, more risk for higher level of parathyroid hormone (>151.1 pg/mL). For other biochemical parameters, we did not detect risk association with *LGALS3* genotypes.

### 3.4. Comparison of Gal-3 Gene Polymorphisms between ESRD Patients with Different Type of Primary Disease

Upon estimation of the risk association between *LGALS3* genotypes and clinical complications, we studied the potential correlation between *LGALS3* genotypes and primary diseases (Table 4). Moreover, binary logistic regression showed that patients who had *LGALS3* rs4644 CC and rs4652 AA genotypes had 3.2- and 10-times-higher risk, respectively, of developing diabetes mellitus (*p* = 0.018; OR = 3.27; CI 1.30–8.60 and *p* = 0.017; OR = 3.27; CI 1.32–8.13, respectively). Additionally, binary logistic regression showed that patients with *LGALS3* rs4644 AA genotype had 6.5-times-higher risk of developing hypertension (*p* = 0.013; OR = 6.46; CI 1.34–31.13). There was no risk association between different type of primary disease and *LGALS3* rs11125 genotypes (*p* > 0.05). 

### 3.5. Frequency of Haplotypes According to Clinical and Biochemical Parameters of ESRD Patients

The CAA haplotype was significantly more common in patients with diabetes (0.796 vs. 0.570; *p* = 0.0029), while the ACA haplotype was more common in patients without diabetes (0.259 vs. 0.093; *p* = 0.0102). Additionally, the CAA haplotype was protective against hypertension (0.691 vs. 0.559; *p* = 0.0469). It has been observed as well that the ACT haplotype was more common in patients with low glomerular filtration (0.167 vs. 0.064; *p* = 0.018). Among patients with normal PTH levels, the CAA haplotype was more frequent than among patients with lower PTH (0.708 vs. 0.547; *p* = 0.0157), while the ACT haplotype was significantly more frequent in patients with low PTH (0.160 vs. 0.066; *p* = 0.0302). Regarding the hemoglobin levels, the CAA haplotype was more common in patients with low hemoglobin levels than in patients with normal hemoglobin levels (0.704 vs. 0.547; *p* = 0.0209), while the ACT haplotype was more frequent in patients with normal hemoglobin (0.160 vs. 0.071; *p* = 0.0488) (Table 5).

## 4. Discussion

The multifactorial etiology of ESRD is well-documented in the previously published literature. In addition, a number of environmental factors, several mutations, and genetic variants were reported to have an association with kidney diseases [19,20,21,22]. In the present study, we investigated for the first time the three SNP polymorphisms of the *LGALS3* gene and its correlation with hemoglobin levels, rate of glomerular filtration, level of parathyroid hormone, diabetes mellitus, and arterial hypertension. First, our findings showed that ESRD patients with the *LGALS3* rs4644 CC and rs4652 AA genotypes and also the CAA haplotype had a higher risk for lower hemoglobin and lower PTH level, and occurrence of diabetes mellitus and arterial hypertension. A recent study indicated that two *LGALS3* gene single-nucleotide polymorphisms (SNPs) (rs4644 and rs4652) are associated with changes in protein levels [7] and may alter the circulating Gal-3 content, but less is known about these genetic variations in relation to CKD or ESRD. However, some limitations should be taken into account when determining serum Gal-3 levels. Namely, repeated measurements at multiple time points can provide better information on the risk of later clinical outcomes. Additionally, genetic variations of rs4644 may affect test results. The variant of rs4652 is associated with Gal-3 expression level, and it was proven that when allele A is changed to C, rs4652 changes threonine to a proline at residue 98, resulting in irregular Gal-3 secretion [23]. 

During a sixteen-year study conducted by Rebholz et al., it was shown that elevated serum Gal-3 values were significantly associated with early and late stages of incident kidney disease in a diverse population with preserved kidney function and without heart failure, particularly among those with hypertension [24].

Our findings are in line with the above study that ESRD patients with rs4644 CC genotypes had a higher risk of occurrence of arterial hypertension with the evidence that individuals with hypertension appeared to be particularly susceptible to the risk of incident CKD in association with elevated Gal-3 levels [24].

Additionally, a recently published study demonstrated that Gal-3 levels, but not *LGALS3* genotypes, were associated with multiple inflammatory marker levels in patients with coronary artery disease. They showed that the A variant of *LGALS3* rs4644 genotypes and the C variant of *LGALS3* 4652 genotypes were associated with relatively low Gal-3 levels [25]. Moreover, the data showed that patients who have *LGALS3* rs4652 pulmonary arterial hypertension and myopathy as predictor variables were statistically significant. Additionally, a decline was observed in the Gal-3 values for the LGALS3 rs4652 A/A, A/C, and C/C genotypes.

Examination of the relationship between biochemical parameters and polymorphisms of the Gal-3 gene showed that ESRD patients with rs4644 CC and rs4652 AA genotypes had 5 and 3 times, respectively, more risk for lower hemoglobin and 3- and 4-times more risk for a lower level of parathyroid hormone.

Both major alleles in rs4644 and rs4652 are in relation with high serum levels of Gal-3, since it is clearly shown that serum Gal-3 level was significantly increased in diabetic and prediabetic patients in comparison to the control group [26]. Therefore, in the continuation of our research, we examined the correlation of these two polymorphisms with the occurrence of diabetes in our population. In accordance with their finding, our results showed that patients who had *LGALS3* rs4644 CC and rs4652 AA genotypes had a 3.2- and 10-times-higher risk of developing diabetes mellitus. 

## 5. Conclusions

In summary, *LGALS3* rs4644 CC and rs4652 AA genotypes were significantly associated with a higher risk for lower hemoglobin, higher level of parathyroid hormone, and also occurrence of diabetes mellitus and arterial hypertension. Determining SNP polymorphisms of the *LGALS3* rs4644 CC and rs4652 AA genotypes may be useful for identifying ESRD patients with poor prognosis, especially for diabetes mellitus and arterial hypertension. Gal-3 may be a target for pharmacotherapy to prevent progression and complications in ESRD patients.

## 6. Limitation of the Study

The first limitation of this study is its modest sample size. Replication of this investigation using a larger sample cohort would improve the strength of the analysis. Second, having in mind that all the patients in the study were ethnically Serbian, caution should be exercised when extrapolating our results to individuals of other ethnicities.

## Figures and Tables

**Figure 1 jcm-11-03874-f001:**
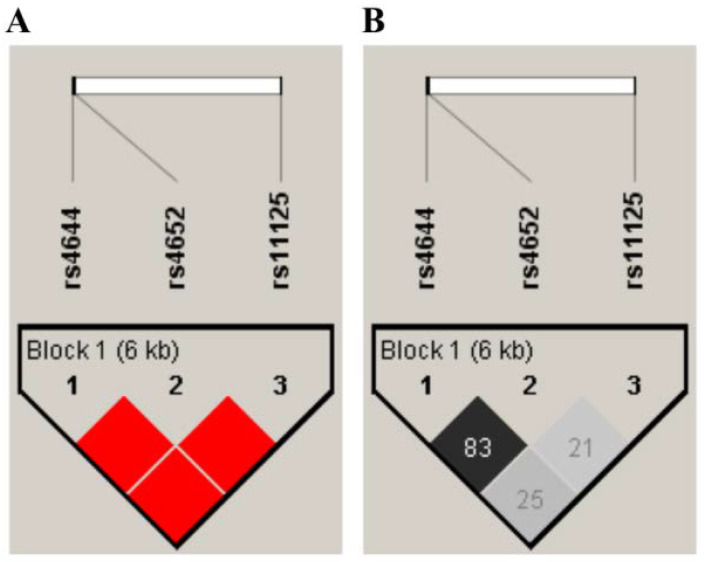
Schematic representation of LD with values of D’ (**A**) and r2 (**B**) among the analyzed polymorphisms of LGALS3 genes (rs4644, rs4652, and rs11125) in patients with renal disease. (**A**) Intense red color indicates a value of D’ = 100 for LOD values > 2. The decrease in the intensity of red is proportional to the decrease in the value of D’. (**B**) The black color is present at the value r^2^ = 100. The intensity of the gray color is proportional to the values of r^2^, where the white color corresponds to the value of r^2^ = 0.

**Figure 2 jcm-11-03874-f002:**
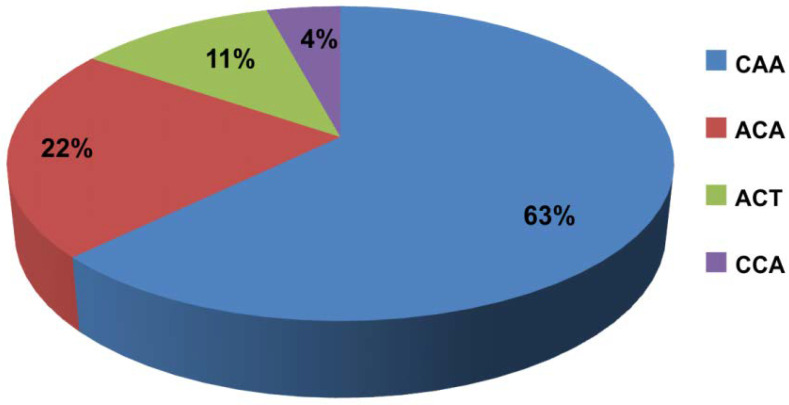
Frequency of haplotypes present in patients.

**Table 1 jcm-11-03874-t001:** Demographic, clinical, and biochemical characteristics of ESRD patients included in the study.

Variables	Normal Value	Mean ± Standard Deviation
Age (years)	>18	63.05 ± 11.74
Length of hemodialysis treatment	>6 months	5.68 ± 5.23 years
Gender	men/women	63/45
Leukocytes	(3.70–10.0 × 10^9^/L)	7.22 ± 3.46
Erythrocytes	(3.86–5.08 × 10¹²/L)	3.74 ± 1.23
Hemoglobin	(110–157 g/L)	102.90 ± 12.32
Platelets	(135–450 × 10^9^/L)	183.86 ± 64.03
Iron	(6.6–26.0 μmol/L)	10.78 ± 5.26
Sodium	(136–145 mmol/L)	138.72 ± 2.98
Potassium	(3.4–4.5 mmol/L)	5.24 ± 0.82
Calcium	(2.02–2.65 mmol/L)	2.27 ± 0.20
Phosphorus	(0.80–1.60 mmol/L)	1.53 ± 0.49
Magnesium	(0.70–1.10 mmol/L)	1.16 ± 0.26
Parathyroid hormone	(120–300 pg/mL)	234.94 ± 271.53
Albumin	(35–52 g/L)	36.96 ± 3.26
Urea before hemodialysis	(3.0–8.0 mmol/L)	21.45± 5.06
Urea after hemodialysis	(3.0–8.0 mmol/L)	7.59 ±2.93
Creatinine before hemodialysis	(49–106 μmol/L)	772.59 ± 512.59
Creatinine after hemodialysis	(49–106 μmol/L)	324.36 ± 106.99
NTproBNP before hemodialysis	(≥7200 pg/mL)	17,491.20 ± 34,341.10
NTproBNP after hemodialysis	(≥7200 pg/mL)	14,575.93 ± 34,478.69
Troponin before hemodialysis	men ≤ 0.03 woman ≤ 0.01	0.02 ± 0.01
Troponin after hemodialysis	men ≤ 0.03 woman ≤ 0.01	0.02 ± 0.01
Glomerular filtration rate	0–15 mL/min	10.94 ± 3.37

Continuous variables are presented as mean ± SD. ESRD, end-stage renal disease; NTproBNP, N-terminal proBrain Natriuretic Peptide.

**Table 2 jcm-11-03874-t002:** Allele frequency of Galectin 3 polymorphisms in ESRD patients and control subjects.

Galectin 3 Polymorphism	Controls *n* = Number (%)	ESRD Patients *n* = Number (%)	*p* Value
**rs4644**			
CC	16 (42.1)	48 (45.3)	0.540
CA	15 (39.5)	46 (43.4)	
AA	7 (18.4)	12 (11.3)	
**rs4452**			
AA	16 (42.1)	44 (41.5)	0.299
AC	12 (31.6)	45 (42.5)	
CC	10 (26.3)	17 (16)	
**rs11125**			
AA	30 (78.9)	84 (79.2)	0.675
AT	8 (21.1)	20 (18.9)	
TT	0 (0)	2 (1.9)	

ESRD, end-stage renal disease.

**Table 3 jcm-11-03874-t003:** Biochemical characteristics of hemodialyzed patients classified according to the Galectin 3 gene polymorphisms.

Clinical Parameters (Normal Values) (Cut-Off Value)	SNP	Genotype	Group 1 * *n* (%)	Group 2 ** *n* (%)	OR; (95% CI)	*p* Value
Hemoglobin (110–157 g/L) (˂103 g/L) **	rs4644	CC	16 (30.2)	29 (59.2)	3.35; (1.48–7.60)	0.006
		CA + AA	37 (69.8)	20 (20.8)		
	rs4652	AA	15 (28.3)	26 (53.1)	2.86; (1.26–6.5)	0.019
		CA + CC	38 (71.7)	23 (46.9)		
	rs11125	AA	38 (71.7)	42 (85.7)	2.37; (0.87–6.43)	0.138
		AT + TT	15 (28.3)	7 (14.3)		
Glomerular filtration rate (≤15 mL/min/1.75 m^2^) (˂11 mL/min/1.75 m^2^) **	rs4644	CC	22 (43.1)	26 (47.3)	1.18; (0.55–2.54)	0.823
		CA + AA	29 (56.9)	29 (52.7)		
	rs4652	AA	20 (39.2)	24 (43.6)	1.2; (0.55–2.60)	0.791
		AC + CC	31 (60.8)	31 (56.4)		
	rs11125	AA	35 (68.6)	49 (89.1)	3.73; (1.33–10.5)	0.018
		AT + TT	16 (31.4)	6 (10.9)		
Parathyroid hormone (120–300 pg/mL) (>151.1 pg/mL) **	rs4644	CC	32 (60.4)	16 (30.2)	3.52; (1.58–7.88)	0.003
		CA + AA	21 (39.6)	37 (69.8)		
	rs4652	AA	30 (56.6)	14 (26.4)	3.63; (1.60–8.23)	0.003
		AC + CC	23 (43.4)	39 (73.6)		
	rs11125	AA	46 (86.8)	38 (71.7)	2.59; (0.96–7.01)	0.090
		AT + TT	7 (13.2)	15 (28.3)		

SNP, single-nucleotide polymorphism; OR, odds ratios; CI, confidence intervals; Group 1 * Low risk; Group 2 ** High risk.

**Table 4 jcm-11-03874-t004:** The frequency distribution and risk association of Galectin 3 genotypes and alleles among the studied groups.

Clinical Parameters	SNP	Genotype	No *n* (%)	Yes *n* (%)	OR; (95% CI)	*p* Value
Diabetes mellitus	rs4644	CC	30 (38.0)	18 (66.7)	3.27; (1.30–8.60)	0.018
		CA + AA	49 (69.8)	9 (20.8)		
	rs4652	AA	27 (34.2)	17 (63.0)	3.27; (1.32–8.13)	0.017
		CA + CC	52 (65.8)	10 (37.0)		
	rs11125	AA	61 (77.2)	23 (85.2)		0.543
		AT + TT	18 (22.8)	4 (14.8)		
Hypertension	rs4644	CC + CA	53 (96.4)	41 (80.4)	6.46; (1.34–31.13)	0.013
		AA	2 (3.6)	10 (19.6)		
	rs4652	AA + AC	50 (90.9)	39 (76.5)		0.078
		CC	5 (9.01)	12 (23.5)		
	rs11125	AA	46 (83.6)	38 (74.5)		0.359
		AT + TT	9 (16.4)	13 (25.5)		
Polycystic kidney disease	rs4644	CC	39 (50.0)	9 (32.1)		0.159
		CA + AA	39 (50.0)	19 (67.8)		
	rs4652	AA	35 (44.9)	9 (32.1)		0.342
		AC + CC	43 (55.1)	19 (67.9)		
	rs11125	AA	61 (78.2)	23 (82.1)		0.862
		AT + TT	17 (21.8)	5 (17.9)		

SNP, single-nucleotide polymorphism; OR, odds ratios; CI, confidence intervals.

**Table 5 jcm-11-03874-t005:** Frequency of haplotypes according to clinical and biochemical parameters of ESRD patients.

Haploypes	With DM	Without DM	*p*	With HTA	Without HTA	*p*	Low GFR	Normal GFR	*p*	High PTH	Normal PTH	*p*	Normal Hgb	Low Hgb	*p*	PKD No	PKD Yes	*p*
CAA	0.796	0.570	0.003	0.559	0.691	0.047	0.636	0.618	0.778	0.547	0.708	0.016	0.547	0.704	0.021	0.641	0.593	0.492
ACA	0.093	0.259	0.010	0.255	0.182	0.197	0.245	0.186	0.296	0.245	0.189	0.317	0.264	0.163	0.080	0.199	0.259	0.282
ACT	0.074	0.127	0.293	0.137	0.091	0.287	0.064	0.167	0.018	0.160	0.066	0.030	0.160	0.071	0.049	0.115	0.111	0.867
CCA	0.037	0.044	0.819	0.049	0.036	0.648	0.055	0.029	0.364	0.047	0.038	0.733	0.028	0.061	0.253	0.045	0.037	0.771

ESRD, end-stage renal disease; DM, diabetes mellitus; PKD, polycystic kidney disease; GFR, glomerular filtration rate; PTH, parathyroid hormone; Hgb, hemoglobin; HTA, hypertension.

## Data Availability

Not applicable.

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
