# Peer review of "Galectin 3 (LGALS3) Gene Polymorphisms Are Associated with Biochemical Parameters and Primary Disease in Patients with End-Stage Renal Disease in Serbian Population"

_jcm, 2022, doi:10.3390/jcm11133874_

Round 1
Reviewer 1 Report
The authors investigated the association of galectin-3 (Gal-3) with biochemical parameters and primary disease in patients with end-stage kidney disease by searching for polymorphisms in the Gal-3 (LGALS3) gene. It has already been reported that Gal-3 is significantly associated with poor prognosis in CKD patients, but as the authors stated, this study is novel in that it reveals an association between SNP polymorphisms in the LGALS3 gene and clinical and biochemical parameters in CKD patients.
I think this study is well conducted, but I have the following concerns:
1. It is difficult to know how to interpret the causal relationship between some LGALS3 gene polymorphisms and clinical and biochemical parameters identified by the authors.
2. For example, re4644 was shown to be associated with diabetes mellitus (DM), does this mean that those with re4644 SNP are more likely to have DM, or that DM patients with re4644 SNP are more likely to develop ESRD?
3. Regardless of underlying disease, renal tissue from patients with end-stage renal failure exhibits common findings such as glomerulosclerosis, interstitial fibrosis, and marked macrophage infiltration. So why do LGALS3 gene polymorphisms differ depending on the underlying disorder in patients with end-stage renal disease who participated in this study?
4. The discussion includes many references and discussions of the literature, but there does not seem to be much discussion based on the author's own data. If the authors had baseline data on serum levels of Gal-3 in patients who participated in this study, it would provide insight into the above question.
Overall, this is an observational study and it may be difficult to discuss the causal relationship between LGALS3 gene polymorphisms and the development of ESRD. I think that the manuscript will be strengthened by adding the above discussion.
Author Response
Kragujevac, 06/16/2022
Dear Ms. Niya Wu,
Many thanks for your patience and the evaluation of our manuscript entitled “Galectin 3 (LGALS3) gene polymorphisms are associated with biochemical parameters and primary disease in patients with end-stage renal disease” (jcm-1765762).
We are thankful for all critical suggestions and comments raised by reviewers and hope that we were able to improve the quality of our manuscript. Please find below our detailed responses and explanations. Please note that all changes are highlighted in red and marked up using the “Track Changes” function.
To Referee 1.
We are very thankful to Referee 1 since the reviewer invested a plethora of time and efforts to improve the quality of our manuscript raising more than constructive questions and suggestions. She/he stated that:
- „It is difficult to know how to interpret the causal relationship between some LGALS3 gene polymorphisms and clinical and biochemical parameters identified by the authors.“
Response: Thank you for the question. In various diseases, it has been shown that different genetic variants of one gene affect the level of proteins encoded by that gene. Regarding the polymorphism of the LGALS3 gene, it has been shown that some but not all variants of this gene affect the final production of serum Gal-3, both at the posttranscription level and at the translation level. Consequently, there is positive correlation between the produced amount of Gal-3 and biochemical and clinical parameters (Constantino, E.G.; Dantas de Lima, C.A.; Vilar, K.M.; Francisco de Nóbrega, M.; Rodrigues de Almeida, A.; Pereira, M.C.; Dantas, A.T.; Guimarães Gonçalves, R.S.; Barreto de Melo Rêgo, M.J.; Pinto Duarte, L.B.; Galdino da Rocha Pitta, M. Genetic variants in LGALS3 are related to lower galectin-3 serum levels and clinical outcomes in systemic sclerosis patients: A case-control study. Autoimmunity. 2021, 54, 187-194.)
- „For example, re4644 was shown to be associated with diabetes mellitus (DM), does this mean that those with re4644 SNP are more likely to have DM, or that DM patients with re4644 SNP are more likely to develop ESRD?.“
Response: Thank you for the question. We hope we were able to answer the raised issue.
We have already mentioned in the Discussions section: „Both major alleles in rs4644 and rs4652 are in relation with high serum levels of Gal-3. Since it is clearly shown that serum Gal-3 level was significantly increased in diabetic and prediabetic patients in comparison to the control group.”
We fully agree with the reviewer's conclusions. Yilmaz and coworkers demonstrated that those with re4644 SNP were particularly susceptible to have DM, and DM patients with re4644 SNP were more likely to develop ESRD (Yilmaz, H.; Cakmak, M.; Inan, O.; Darcin, T.; Akcay, A. Increased levels of galectin-3 were associated with prediabetes and diabetes: new risk factor? J. Endocrinol. Invest. 2015, 38, 527-533.). Our results are in accordance with these findings, as clearly demonstrated in Table 4.
- „Regardless of underlying disease, renal tissue from patients with end-stage renal failure exhibits common findings such as glomerulosclerosis, interstitial fibrosis, and marked macrophage infiltration. So why do LGALS3 gene polymorphisms differ depending on the underlying disorder in patients with end-stage renal disease who participated in this study?“
Response: Thank you for the question. We fully agree with the reviewer that renal tissue from patients with end-stage renal failure exhibits common pathohistological changes such as glomerulosclerosis, interstitial fibrosis, and marked macrophage infiltration, however, despite the same PH changes and the same diagnosis (ESRD), these patients have different cardiovascular complications that can potentially be life-threatening. It is known that the LGALS3 gene polymorphisms has a significant correlation with the development of cardiovascular disease, for this reason, determining the LGALS3 gene polymorphisms would potentially provide a subtle difference between patients with “sensitive” haplotype and those who do not have it, in the context of the risk for cardiovascular complications.
- „The discussion includes many references and discussions of the literature, but there does not seem to be much discussion based on the author's own data. If the authors had baseline data on serum levels of Gal-3 in patients who participated in this study, it would provide insight into the above question.“
Response: This is a very helpful advice. As we have already mentioned in the Discussions section: „However, some limitations should be taken into account when determining serum Gal-3 levels. Namely, repeated measurements, measurements at multiple time points, can provide better information on the risk of later clinical outcomes. Additionally, genetic variations on rs4644 may affect test results.” Additionally, having in mind that pathological processes of fibrosis and inflammation in ESRD patients are ended, serum Gal-3 level would not be important data for these patients. However, determining serum Gal-3 levels may be useful diagnostic information for patients with early CKD as well as during the monitoring of disease progression.
All suggestions are accepted and the manuscript was modified in accordance with them, highlighted in the manuscript by using the option review “track changes”.

Reviewer 2 Report
The manuscript "Galectin 3 (LGALS3) gene polymorphisms are associated with biochemical parameters and primary disease in patients with end-stage renal disease" show an association between LGALS3 rs4644 and rs4652 gene polymorphism and chronic kidney disease. The manuscript is well written and findings are of relevance. Some considerations below:
1. The abstract section only focuses on the results regarding galectin 3 gene polymorphisms and its association with clinical and biochemical parameters. Results regarding the frequency of haplotypes should be also mentioned in the abstract section.
2. The manuscript could benefit from the investigation of circulating galectin-3 levels in ESRD patients.
3. The discussion section should be improved to answer some questions: - What is the association between the galectin-3 physiological functions and LGALS3 genotypes rs4644, rs4652 and rs11125? - What is the association between the different LGALS3 genotypes (rs4644, rs4652 and rs11125)? . What are the limitations of this study? In my point go view, this study has some limitations. The first limitation is its modest sample size. Replication of this investigation using a larger sample cohort would improve the strength of the analysis. In addition, the cross-sectional design of this study limits our ability to infer causal relationships between LGALS3 genotypes, galectin-3 levels, and inflammatory marker levels. Finally, all the patients in the study were ethnically Chinese; therefore, caution should be exercised when extrapolating our results to other ethnic groups. 4. CKD is described as both Chronic Kidney Disease and as Chronic renal failure, please revise.
Author Response
Kragujevac, 06/16/2022
Dear Ms. Niya Wu,
Many thanks for your patience and the evaluation of our manuscript entitled “Galectin 3 (LGALS3) gene polymorphisms are associated with biochemical parameters and primary disease in patients with end-stage renal disease” (jcm-1765762).
We are thankful for all critical suggestions and comments raised by reviewers and hope that we were able to improve the quality of our manuscript. Please find below our detailed responses and explanations. Please note that all changes are highlighted in red and marked up using the “Track Changes” function.
To Referee 2:
We are very thankful to Referee 2 since the reviewer invested a plethora of time and efforts to improve the quality of our manuscript raising more than constructive questions and suggestions. She/he stated that:
- „The abstract section only focuses on the results regarding galectin 3 gene polymorphisms and its association with clinical and biochemical parameters. Results regarding the frequency of haplotypes should be also mentioned in the abstract section“.
Response: Thank you for the sharp observation. In the new version of the manuscript, we modified abstract section according your suggestions.
New abstract: Galectin 3 plays a significant role in the development of chronic renal failure, particulary end-stage of renal disease (ESRD). The aim of our study was to investigate the association between Gal-3 and biochemical parameters and primary disease in ESRD patients, by exploring the polymorphisms LGALS3 rs4644, rs4652, and rs11125. A total of 108 ESRD patients and 38 healthy controls were enrolled in the study. Genotyping of LGALS3 gene rs4644, rs4652 and rs11125 polymorphisms was performed by polymerase chain reaction–restriction fragment-length polymorphism (PCR–RFLP). By multivariate logistic regression analysis, we found that LGALS3 rs4644 CC and rs4652 AA genotypes were significantly associated with higher risk for lower hemoglobin, higher level of parathyroid hormone, and also occurrence diabetes mellitus and arterial hypertension. The CAA haplotype was significantly more common in patients with diabetes, low hemoglobin level, and normal PTH level. It has been observed as well that ACT haplotype was more common in patients with low glomerular filtration, low PTH and normal hemoglobin level. We found that the LGALS3 rs4644 and rs4652 gene polymorphism may be involved in the pathogenesis and appearance of complications in ESRD patients and thus could be considered a new genetic risk factor in this population.
- „The manuscript could benefit from the investigation of circulating galectin-3 levels in ESRD patients.“
Response: Thank you for this question. We appreciate the referee's consideration, however, in our response to the Referee 1, we provided justification for these analisys.
As we have already mentioned in the Discussions section: „However, some limitations should be taken into account when determining serum Gal-3 levels. Namely, repeated measurements, measurements at multiple time points, can provide better information on the risk of later clinical outcomes. Additionally, genetic variations on rs4644 may affect test results.” Additionally, having in mind that pathological processes of fibrosis and inflammation in ESRD patients are ended, serum Gal-3 level would not be important data for these patients. However, determining serum Gal-3 levels may be useful diagnostic information for patients with early CKD as well as during the monitoring of disease progression.
- „The discussion section should be improved to answer some questions: What is the association between the galectin-3 physiological functions and LGALS3 genotypes rs4644, rs4652 and rs11125? What is the association between the different LGALS3 genotypes (rs4644, rs4652 and rs11125)? “
Response: Thank you for your kind suggestion. As we have already mentioned in the Discussions section: „A recent study has indicated that two LGALS3 gene single nucleotide polymorphisms (SNPs) (rs4644 and rs4652) are associated with changes in protein levels and may alter the circulating Gal-3 content in a relatively healthy population [De Boer, R.A.; Verweij, N.; van Veldhuisen, D.J.; Westra, H.J.; Bakker, S.J.; Gansevoort, R.T.; Kobold, A.C.; van Gilst, W.H.; Franke, L.; Leach, I.M.; van der Harst, P. A genome-wide association study of circulating galectin-3. PLoS One. 2012, 7:e47385.] “.
Our additional explanation for your questions: In another study, significant associations between these LGALS3 SNP genotypes and galectin-3 levels were evident. The minor allele of both rs4644 and rs4652 were significantly associated with lower galectin-3 level (0.68 ng/ml and 3.71 ng/ml, respectively) [Liao YH, Teng MS, Juang JJ, Chiang FT, Er LK, Wu S, Ko YL. Genetic determinants of circulating galectin-3 levels in patients with coronary artery disease. Mol Genet Genomic Med. 2020 Sep;8(9):e1370]. In particular, it was demonstrated that rs4644 CC genotype and rs4652 AA genotype were associated with higher galectin-3 levels. Gal-3 is a well-known proinflammatory marker that may significantly affect pathogenesis of disease.“
- „What are the limitations of this study? In my point go view, this study has some limitations. The first limitation is its modest sample size. Replication of this investigation using a larger sample cohort would improve the strength of the analysis. “
Response: Thank you for your suggestion. The revised version of the manuscript contains more clear statement about limitations of our study. In order to fulfil request raised by reviewer, we added the following paragraph.
Limitation of the study
First limitation of this study is its modest sample size. Replication of this investigation using a larger sample cohort would improve the strength of the analysis. Second, having in mind that all the patients in the study were ethnically Serbian; caution should be exercised when extrapolating our results to individuals of other ethnicities.
- „In addition, the cross-sectional design of this study limits our ability to infer causal relationships between LGALS3 genotypes, galectin-3 levels, and inflammatory marker levels. Finally, all the patients in the study were ethnically Chinese; therefore, caution should be exercised when extrapolating our results to other ethnic groups.“
Response: We are very thankful for this critical comment. We believe that Referee 2 accidentally made a mistake, because in our study we examine the LGLA3S polymorphism in the Serbian population (not in Chinese as reviewer stated). In revised version of the manuscript we added a new paragraph regarding limitation of our study.
Limitation of the study
First limitation of this study is its modest sample size. Replication of this investigation using a larger sample cohort would improve the strength of the analysis. Second, having in mind that all the patients in the study were ethnically Serbian; caution should be exercised when extrapolating our results to individuals of other ethnicities.
Moreover, according to the reviewers suggestion, in order to improve the quality of our manuscript we modified the title, since all the patients in the study were ethnically Serbian.
New title: Galectin 3 (LGALS3) gene polymorphisms are associated with biochemical parameters and primary disease in patients with end-stage renal disease in Serbian population
- „CKD is described as both Chronic Kidney Disease and as Chronic renal failure, please revise.“
Response: Thank you for your suggestion, this is now corrected. Chronic Kidney Disease is described as CKD throughout revised manuscript.
All suggestions are accepted and the manuscript were modified in accordance with them, highlighted in the manuscript by using the option review “track changes”.

Round 2
Reviewer 1 Report
The authors have properly responded to my comments. I look forward to the next study that addresses what the authors have described in Limitation.